# Well-Being in the Time of Corona: Associations of Nearby Greenery with Mental Well-Being during COVID-19 in The Netherlands

**Ralitsa Shentova** [1,*] , **Sjerp de Vries** [2] **and Jana Verboom** [1]

1   Environmental Systems Analysis, Wageningen University & Research, 6708 PB Wageningen, The Netherlands
2   Wageningen Environmental Research/Cultural Geography, Wageningen University & Research,
    6708 PB Wageningen, The Netherlands
*   Correspondence: ralitsa.shentova@gmail.com

**Abstract:** Nature's mental health benefits are well-established in the literature, but there is little research on which types and characteristics of urban greenery are most relevant for mental well-being in general, and during the COVID-19 pandemic in particular. This study examined the link between having a (green) garden or a green view from the main window of the home, as well as the perceived quantity and quality of neighbourhood green areas and streetscape greenery, and the self-reported change in mental well-being since the onset of the COVID-19 pandemic. Adults residing in the Netherlands (N = 521, 67% female) completed an online survey in December 2020 and January 2021. It included items on the frequency of contact with the aforementioned outdoor spaces, as well as their quantity, natural features, and quality. Hierarchical regression analyses revealed that the quantity of the greenery mattered, but the quality was more strongly associated with well-being. In particular, well-maintained, attractive, and varied streetscape greenery was just as relevant as a garden with diverse plants. This beneficial association between streetscape greenery and mental well-being was stronger for female participants. Understanding the benefits of the different types and characteristics of urban greenery, and who they are most relevant for, can assist policymakers and planners in designing cities that promote health and resilience.

**Keywords:** urban planning; multilevel analysis; mental health; green infrastructure; public health

## 1. Introduction

In response to the COVID-19 pandemic that hit the world in 2020, governments implemented containment measures, such as social distancing, stay-at-home orders, and the prohibition of public space use. While successful in slowing the spread of the virus, these measures, along with the situation in general, have had impacts on psychological well-being [1]. The ongoing pandemic has been associated with deteriorated mental health compared with pre-COVID-19 levels [2,3], exemplified by widespread symptoms of anxiety, depression [4], and stress [5]. However, research also suggests that more greenery in the residential environment may mitigate these negative mental health impacts, as the following paragraphs will elaborate.

Due to their population size and global and local interconnectivity, urban areas have become epicentres of the pandemic [6]. In combination with the projection that 70% of the world's population is expected to be living in cities by 2050 [7], this highlights the importance of creating city environments that support their residents' well-being. Pre-pandemic research has indicated that nearby green areas and natural elements (e.g., street trees) are important for the mental health and well-being of urban dwellers [8,9]. Now, their importance is increasingly recognised and included in COVID-19 recovery plans [10,11]. The outdoors in general, and even window views of greenery, helped people cope with the negative effects of COVID-19 restrictions, and were associated with better mental health

and well-being [12]. The pandemic also saw an increase in visits to urban and peri-urban green areas (e.g., in the USA) [13], as well as small publicly accessible urban gardens nearby (e.g., in Italy) or tree-lined streets (e.g., in Spain, Israel) [14]. In order to create cities that are conducive to good mental health for their residents, it is important to understand what types of green areas and natural elements are most relevant for well-being, and why.

Different types of greenery may have different effects on urban dwellers. Research so far has not been focused on examining which are the most relevant types and characteristics of urban greenery for mental health [15]. Some studies have drawn distinctions between streetscape greenery and public green areas [16]. The former refers to natural elements visible in the street, such as trees, hedges, verges, and private front gardens, while the latter includes parks, publicly accessible gardens, and grassy playing fields. Higher streetscape greenery quality and quantity and green area quantity have been associated with better mental health [16]. Private gardens are another type of urban greenery that many city residents are likely to experience in their daily lives [17]. Positive associations have been found between mental well-being and having a garden [18–20]. In this study, the greenery visible from a window was considered as yet another type of urban greenery, although it overlapped with the aforementioned types to some extent. Studies have shown that a green window view is also associated with well-being [21–24], so it is important to examine how its impact compares with other ways of experiencing green spaces and natural elements. Urban greenery can vary greatly in size and characteristics. Being specific regarding its types and features and their associated benefits can enable embedding it in urban design in the most resource-efficient and effective ways to promote well-being.

Research on urban greenery's impacts on well-being so far has largely focused on the exposure, amount, and proximity of green spaces (e.g., parks) or smaller natural elements (e.g., trees, hedges) [15]. Indeed, the frequency of contact and time spent in such areas play a role in the size of the effect [25–27]. Studies have also shown that the type of contact with a green area also matters for its effects—e.g., more passive contact, such as a view from a balcony, may be different from more active contact, such as gardening [28]. Gardening is one of the most common ways people interact with natural elements and is an activity that has been associated with improved mental health [17,29,30]. Thus, in addition to looking at the different types and characteristics of urban greenery, it is also relevant to incorporate variables on the exposure and type of contact into analyses of its effects on well-being. This will help gain a better understanding the path(s) between the greenery and its benefits.

To incorporate the aforementioned variables, this study used the framework proposed by Bratman et al. (2019). It is a stepwise model presenting a way to examine the pathway from the environment to mental health. According to this model, the natural features or characteristics of an area (e.g., greenness, size) have effects on mental health through exposure, referring to the amount of contact (e.g., duration and frequency of contact), and engagement (i.e., types and intensity of interactions during the exposure) [31]. This model allows for the inclusion of relevant elements of the natural environment, as well as how—through what exposure and engagement—those may lead to certain benefits.

Lastly, the relation between green spaces and health may vary by socioeconomic status, age, employment status, gender, and nature connectedness [15]. This emphasises the need to examine not only the characteristics and the use of nearby urban greenery, but also for whom they are most relevant, to identify the most effective ways to promote the mental health benefits for specific population segments.

This research aimed to assess to what extent variations in the impact of COVID-19 on well-being in 2020 can be explained by differences in access to and contact with different types of greenery in the residential environment. This was achieved by first looking at the associations between self-reported changes in mental health since the onset of the pandemic and several types of greenery—namely, private gardens, public green areas, and streetscape greenery through their essential characteristics (as perceived by participants), and green window views—and the exposure to and engagement with them as potential mediators.

This was followed by a comparison between these types and an exploratory moderation analysis with age, gender, nature connectedness, and level of education.

## 2. Materials and Methods

### 2.1. Setting

This study was conducted in the Netherlands, where 92% of the population resides in urban areas [32]. For the majority of 2020, the country followed "intelligent lockdown measures" which entailed the cancellation of events with over 100 people, a shift to online education, and the requirement for businesses to operate with a 1.5 m distance regulation and sanitation measures in place [33]. Measures were reduced in summer [34], but due an increase in cases in autumn, mask-wearing became compulsory indoors as of 1 December and a hard lockdown came into effect as of 15 December. This meant all venues and non-essential shops were closed and non-medical contact-based work was put on hold, alongside continued online education, working from home, and suggestions for social distancing [35]. Research showed that during the pandemic, Dutch residents travelled less [36], and urban dwellers, in particular, experienced decreased well-being [33].

### 2.2. Study Design

Data were gathered using an online questionnaire. The questionnaire started by asking participants about private outdoor areas and neighbourhood greenery (residential greenery), a well-being assessment, a nature connectedness scale, and lastly, sociodemographic and household questions. The following paragraphs elaborate on the contents of each section. The entire questionnaire can be found in the Supplementary Materials.

#### 2.2.1. Residential Greenery

In terms of residential greenery, participants were asked about types of outdoor areas attached to their homes (e.g., garden, balcony), if any. Then, the questions on neighbourhood greenery drew a distinction between streetscape greenery and public green areas. The final question was about the greenness of the view from the window of their living room. Questions on the characteristics of the aforementioned spaces used 7-point Likert scales, with the exception of the size of the private garden, which participants were asked to estimate in square meters. For example, "How much vegetation is there in this space (e.g., trees, grass, shrubs)?" on a scale from "none at all" to "quite a lot" to measure greenness, or "Throughout the year in this space, do you see . . . " with "very few bird species" to "very many bird species" to measure bird diversity.

For both private (i.e., outdoor spaces attached to the home) and neighbourhood green areas, the exposure was measured with the frequency of contact, e.g., "How often have you been here in your spare time this year (2020)?" ("almost never" to "very often"). The activity of gardening in a private outdoor space was included with the following question: "How much of the time that you spent in this space did you engage in gardening?" ("almost all the time" to "less than half the time").

#### 2.2.2. Well-Being Assessment

Following the questions on the outdoor areas, changes in mental well-being were measured with an adapted version of the 12-item General Health Questionnaire (GHQ-12). GHQ-12 is a widely used unidimensional measure for psychological distress in non-clinical samples and has been applied in different cultures [37]. To fit our research purposes, the main question was rephrased from "Have you recently . . . " to "Since the COVID-19 pandemic reached The Netherlands (consider March 2020 when measures began to be implemented nationally), have you . . . ". The sub-questions then ask about various symptoms and behaviours experienced recently (e.g., " . . . lost much sleep over worry?") and—as in the original GHQ—each item is to be rated on a four-point Likert scale, e.g., "not at all" to "much more than usual". The word "usual" on the scale labels was changed

to "before", referring to before the onset of the pandemic. Thus, an adapted GHQ was used to measure the self-reported change in well-being since the onset of the pandemic.

### 2.2.3. Covariates

Socio-demographic and other variables are known to affect the association between well-being and outdoor spaces [15]. This study examined the effects of some of those, namely age (measured in three age groups: 18–34, 35–64, and 65+), gender (male, female, or other gender identity), highest level of attained education (containing nine response options to encompass the Dutch educational landscape), and nature connectedness. The last variable was measured with the six-item Nature Relatedness scale (NR-6). Strong convergence was found among different measures for connectedness with nature, such as the NR, COM (Commitment to Nature), CTN (Connectedness to Nature), and CWN (Connectivity with Nature) scales, so they are considered as markers of the same construct [38]. NR-6 contains statements (e.g., "My connection to nature and the environment is a part of my spirituality") that the respondents are asked to rate on a five-point scale of how much they agree with each (i.e., "disagree", "slightly disagree", "neutral", "slightly agree", or "agree"). It has shown good internal consistency and temporal stability, and has also predicted nature contact and happiness, which is related to mental well-being [39]. The nature connectedness score of respondents was calculated as the mean of the item scores.

### 2.2.4. Dissemination

The questionnaire was hosted on the Qualtrics survey platform and was available in Dutch as well as English. It was disseminated between 14 December 2020 and 25 January 2021. It was shared on social media, neighbourhood and expat forums, as well as in newsletters of nature-related organizations and on the website of Wageningen Environmental Research. Thus, the present study used a convenience sample, which depended on the participants' self-enrolment.

### *2.3. Data Analysis*
### 2.3.1. Statistical Analyses

Data were analysed using IBM SPSS Statistics, version 28. Independent samples *t*-tests were used to examine the differences in well-being between people who had an outdoor space attached to their home (and in particular a garden) and those who did not. Correlations were used to examine if there were significant relationships between the different independent variables (IVs). Linear regression analyses were used to study the predictive power of the various IVs and to quantify their relationship with the dependent variable (change in well-being). These were performed using a hierarchical model comparison, as this allowed for the statistical examination of the contribution of new variables within the models predicting well-being. They followed the logic of first adding the more general characteristics, i.e., how much greenery is present (e.g., size, quantity), and then adding the more specific ones in the following steps, i.e., what is that greenery like (e.g., abundance and diversity of different plants). Each model builds on the findings of the previous ones by retaining the significant predictors identified in them. The predictors that remained significant in the last steps of the regression analyses were considered most relevant for well-being in this study and were used in the following analyses for mediation and moderation.

To shed more light on the pathway between environment and health, exposure and engagement measurements, i.e., frequency of contact and time spent engaging in gardening, were tested as mediators in the previously identified associations. This was achieved through bootstrapping using the PROCESS macro (version 4.1) for SPSS, Model 4. The analysis utilised 5000 bootstrap samples and generated 95% confidence intervals (CI) of the indirect effects of the mediators. CIs that did not include zero were considered statistically significant [40].

We also explored the moderating effects of age, gender, education, and nature connectedness in the relationship between the most relevant predictors and well-being. This was

achieved by adding multiplicative interaction terms with a dummy variable to hierarchical regressions.

### 2.3.2. Data Preparation

Based on the number of responses in the different categories and to keep the exploratory analyses simple, some moderators were dichotomised before creating the interaction terms. For education, the responses were divided into higher, which included both university and university of applied science (79% of respondents), and lower, which included all the other types (with values for higher levels = 1, and lower levels = 0). For gender, only the given responses "male" or "female" were used (N = 512; female = 1, male = 0). The response options for age consisted of three age groups. Research focusing on older adults as a specific population group suggests that they use and experience outdoor spaces differently [15]. Taking this into account, we dichotomised the age variable to 65+ (29% of respondents; 1) and others (0). Lastly, the measure for nature relatedness (Cronbach's $\alpha$ = 0.82) was mean-centred. For the moderation analyses, the previously identified most relevant predictors were also mean-centred before creating the multiplicative interaction terms.

The variable for the size of the private outdoor area was logarithmically transformed to normalise the distribution. The quality of streetscape greenery was assessed by three items: variety, maintenance, and attractiveness (Cronbach's $\alpha$ = 0.84). The quality of green areas was assessed by five items: safety, variety, maintenance, attractiveness, and presence of amenities (Cronbach's $\alpha$ = 0.85).

GHQ-12 scores were reverse-coded so that higher scores illustrated a less negative or positive change in well-being since the onset of the COVID-19 pandemic (later also referred to as a "change in well-being"). Then, the mean scores were calculated. Thus, a score of 1 indicates maximally worsened well-being, and a score of 4 indicates the greatest improvement in well-being (Cronbach's $\alpha$ = 0.89).

More details on data preparation as well as additional results and correlation tables between the variables of interest can be found in the Supplementary Materials.

### 2.4. Participants

A total of 521 responses were used for analysis. Table 1 displays the type of outdoor areas respondents had attached to their home and the main socio-demographic characteristics of the respondents, along with their nature relatedness (NR) and their changes in mental well-being associated with the COVID-19 pandemic and measures (GHQ-12 score).

**Table 1.** Types of outdoor areas attached to the home, socio-demographic characteristics, GHQ score, and nature relatedness (with SD) of the study sample (N = 521).

| Variable | Total |
|---|---|
| **Outdoor areas attached to the home * (%)** | |
| Private garden | 80.8 |
| Balcony | 19.8 |
| Roof terrace | 6.5 |
| Gallery/balcony shared with other households | 3.6 |
| Closed courtyard shared with other households | 3.5 |
| Private courtyard/patio | 3.5 |
| Public courtyard | 3.6 |
| Other | 3.6 |
| None | 3.1 |

**Table 1.** *Cont.*

| Variable | Total |
|---|---|
| **Socio-demographic characteristics (%)** | |
| Female | 66.6 |
| Aged 18–34 | 17.27 |
| Aged 35–64 | 53.55 |
| Aged 65+ | 29.17 |
| **Level of education (%)** | |
| Lower level of education | 20.7 |
| Higher level of education | 79.3 |
| **Nature relatedness (NR-6 score)** | 4.06 (0.69) |
| **Well-being change (GHQ-12 score)** | 2.96 (0.48) |

* Note: respondents who had an outdoor space attached to the home (N = 505) could indicate multiple options.

## 3. Results

### 3.1. Private Outdoor Areas

COVID-19 orders and restrictions meant that more people were spending more time at home compared to prior to the pandemic. This is why we started by looking at the well-being changes associated with having outdoor spaces attached to the home and their essential characteristics, and then moved outwards to public greenery. An independent samples *t*-test analysis showed a significant difference in the change in well-being between people who had an outdoor space attached to their home (M = 2.97, SD = 0.47) and those who did not (M = 2.73, SD = 0.68); t(519) = 1.94, $p < 0.05$. This indicates that having a private outdoor space was associated with a lower negative impact on well-being during the pandemic and its accompanying measures in 2020.

Since having a private garden was the most common type of outdoor space, and likely to be considerably larger and contain more greenery (in absolute terms) than the other types of outdoor space, people having a garden were compared to those who only had another type of outdoor space. This second test showed a significant difference in the GHQ scores between people who had a garden (M = 2.99, SD = 0.46) and those who did not, but did have another type of outdoor space attached to their home (M = 2.84, SD = 0.50); t(503) = −2.63, $p < 0.01$. Thus, having a private outdoor space seemed to mitigate the impacts of COVID-19, but a garden helped more than the other (usually smaller) types of outdoor spaces (e.g., balconies). Therefore, the following analyses focus on the subset of participants having a private garden.

### 3.2. Private Garden

#### 3.2.1. Characteristics

The hierarchical regression analysis models presented in Table 2 illustrate which features of gardens are relevant predictors for well-being. For Step 1 in both Models 1 and 2, only garden size was a predictor. The results showed that an increase in size was associated with a less negative change in well-being. Adding the garden's greenness (Model 1) did not improve the model. Then, in Step 2 of Model 2, the abundance of grass, flowers/herbs, shrubs, and trees were added, which improved the model as a whole. The results indicated that only flower/herb abundance was significantly associated with a less negative impact on mental well-being, along with size. This variable was added to garden size in Step 1 of Model 3, followed by the variable measuring the diversity of plants in Step 2, and the one on the diversity of birds in Step 3. The model improved in Step 2, and the flower/herb abundance was no longer significant. Thus, the results indicate that of all the examined characteristics, plant diversity and garden size were the most important predictors of a less negative change in mental well-being.

**Table 2.** Hierarchical regression analysis results for variation in well-being with private garden characteristics.

| Predictors (β Values) | Model 1 | | Model 2 | | Model 3 | |
|---|---|---|---|---|---|---|
| | Step 1 | Step 2 | Step 2 | Step 1 | Step 2 | Step 3 |
| Size | 0.179 *** | 0.143 ** | 0.186 ** | 0.158 ** | 0.120 * | 0.127 * |
| Greenness | | 0.085 | | | | |
| Abundance of grass | | | −0.032 | | | |
| Abundance of flowers/herbs | | | 0.123 * | 0.127 ** | 0.062 | 0.063 |
| Abundance of shrubs | | | 0.066 | | | |
| Abundance of trees | | | −0.089 | | | |
| Plant diversity | | | | | 0.132 * | 0.148 * |
| Bird diversity | | | | | | −0.032 |
| Adj. $R^2$ | 0.030 | 0.033 | 0.043 | 0.043 | 0.052 | 0.050 |
| $\Delta R^2$ | 0.032 | 0.006 | 0.23 | 0.048 | 0.011 | 0.001 |
| $\Delta F$ | 13.77 *** | 2.56 | 2.44 * | 10.40 *** | 4.80 * | 0.28 |

\*: $p < 0.05$, \*\*: $p < 0.01$, \*\*\*: $p < 0.001$. Dependent variable: GHQ (well-being change). N = 415.

### 3.2.2. Exposure and Engagement as Mediators

Exposure and engagement were examined for their mediating effects. Exposure was measured through how often respondents were in the garden in their spare time, i.e., frequency of contact. In terms of engagement, gardening was included, measured in relation to the total time spent in the garden, i.e., "How much of the time you spent in this space did you engage in gardening". Since size and plant diversity were identified as the most relevant for improved well-being in the previous section, they were used as IVs for the two mediation analyses. By examining the indirect paths in more detail, the results showed that both IVs were predictors for both mediators. The mediators, however, did not predict the GHQ scores. Looking at the total indirect path, the CIs for the effects of the exposure and engagement variables included zero in both cases, which indicated that no significant mediation was occurring. The PROCESS output can be found in the Supplementary Materials.

### 3.3. Public Greenery

#### 3.3.1. Quantity or Quality, and Types

Next, we examined the effect of public greenery on the changes in well-being for all participants, looking at quantity and then quality (Table 3). Starting with public green areas (Model 4), the regression analyses illustrated that the quantity was relevant, but adding quality made quantity redundant while improving the model. However, the overall model fit was still low, pointing to confounding factors.

**Table 3.** Regression analysis results for variation in well-being with public greenery types and their characteristics.

| Predictors (β Values) | Model 4 | | Model 5 | | Model 6 |
|---|---|---|---|---|---|
| | Step 1 | Step 2 | Step 1 | Step 2 | - |
| Public green areas—amount | 0.093 * | 0.050 | | | |
| Public green areas—quality | | 0.106 * | | | 0.042 |
| Streetscape greenery—amount | | | 0.103 * | 0.008 | |
| Streetscape greenery—quality | | | | 0.166 ** | 0.146 ** |
| Adj. $R^2$ | 0.007 | 0.014 | 0.009 | 0.025 | 0.026 |
| $\Delta R^2$ | 0.009 | 0.009 | 0.011 | 0.018 | 0.030 |
| $\Delta F$ | 4.52 * | 4.89 * | 5.55 * | 9.84 ** | 8.05 *** |

\*: $p < 0.05$, \*\*: $p < 0.01$, \*\*\*: $p < 0.001$. Dependent variable: GHQ (well-being change). N = 520.

Similar to public green areas, although the quantity of streetscape greenery was also a significant predictor, adding quality made it redundant and improved the model further

(Model 5). In the final model, the relative well-being impacts of the quality of public green areas and that of streetscape greenery were examined, as those were shown to be most significant. The results pointed to the streetscape greenery's quality as a stronger predictor for better changes in well-being, as only that variable remained significant (Model 6).

### 3.3.2. Exposure as a Mediator

For the mediation analysis, the quality of public green areas was examined as an IV, and the frequency of visits to them was examined as a mediator. The IV was a predictor for the mediator, but the mediator was not one for well-being. The results showed that the CIs for the indirect effect included zero, hence not supporting mediation.

### 3.4. Relative Importance of Different Types of Greenery

Thus far, the data indicated that for those participants who have a private garden, its size and plant diversity are the most relevant indicators for changes in well-being, and that for all participants, the quality of the streetscape greenery was most relevant. The following analyses were used to assess the relative importance of the different types of greenery for well-being, for participants who have a garden. Firstly, a correlation analysis was used to assess how related the predictors identified in earlier models were, now including window view greenness (Table 4). It indicated low positive correlations between all predictors except the quality of streetscape greenery and garden size. As window view greenness showed a significant positive correlation with well-being (Table 4), it was also included as a predictor in the following model.

**Table 4.** Correlations between main predictors for changes in well-being.

| Variable | M | SD | 1. | 2. | 3. | 4. |
|---|---|---|---|---|---|---|
| 1. Garden size | 4.18 | 0.61 | - | | | |
| 2. Garden plant diversity | 5.60 | 1.34 | 0.367 ** | - | | |
| 3. Streetscape greenery quality | 4.34 | 1.37 | 0.056 | 0.098 * | - | |
| 4. Window view greenness | 5.08 | 1.79 | 0.376 ** | 0.408 ** | 0.315 ** | - |
| 5. Change in well-being | 2.99 | 0.46 | 0.179 ** | 0.193 ** | 0.178 ** | 0.177 ** |

*: $p < 0.05$, **: $p < 0.01$. N = 416.

The hierarchical regression Model 7 (Table 5) draws a comparison between the different types of greenery, examining the relative strength of their associations with changes in the well-being of the participants who have gardens. The order of entry started nearer to the home, in particular from inside the home, and then moved further away. The results indicated that on its own, a greener view was associated with better well-being (Step 1). However, it became and remained redundant with the addition of private garden size, plant diversity, and the quality of streetscape greenery in Steps 2 and 3. In the end, the results indicated that garden plant diversity and the quality of streetscape greenery were the most relevant predictors for a less negative change in well-being.

**Table 5.** Hierarchical regression analysis results for variation in well-being with the relevant features of different types of greenery.

| Predictors (β Values) | Model 7 | | |
|---|---|---|---|
| | Step 1 | Step 2 | Step 3 |
| Window view greenness | 0.186 *** | 0.095 | 0.045 |
| Garden size | | 0.093 | 0.103 |
| Garden plant diversity | | 0.137 * | 0.140 ** |
| Streetscape greenery quality | | | 0.142 ** |
| Adj. $R^2$ | 0.032 | 0.056 | 0.072 |
| $\Delta R^2$ | 0.034 | 0.029 | 0.018 |
| $\Delta F$ | 14.75 *** | 6.33 ** | 8.07 ** |

*: $p < 0.05$, **: $p < 0.01$, ***: $p < 0.001$. Dependent variable: GHQ (well-being change). N = 416.

### 3.5. Moderation

Lastly, moderation analyses were conducted to examine how other variables affect the previously identified significant relationships (Table 6). In Step 1, garden plant diversity, the quality of the streetscape greenery, and the previously identified potential moderators were included. In Step 2, the interaction terms were added for education (Model 8a), age (8b), nature relatedness (8c), and gender (8d), respectively.

**Table 6.** Hierarchical regression results for moderation models with education, age, nature relatedness (NR), and gender.

| Predictors (β Values) | Model 8a | | Model 8b | | Model 8c | | Model 8d | |
|---|---|---|---|---|---|---|---|---|
| | Step 1 | Step 2 | Step 1 | Step 2 | Step 1 | Step 2 | Step 1 | Step 2 |
| Garden plant diversity | 0.178 *** | 0.214 * | 0.170 ** | 0.167 ** | 0.060 *** | 0.060 *** | 0.181 *** | 0.288 ** |
| Streetscape greenery quality | 0.064 *** | 0.236 * | 0.154 ** | 0.157 ** | 0.054 *** | 0.053 ** | 0.169 *** | −0.038 |
| Education | 0.053 | 0.163 | | | | | | |
| Education × plant diversity | | −0.087 | | | | | | |
| Education × streetscape quality | | −0.148 | | | | | | |
| Age | | | 0.112 * | 0.283 | | | | |
| Age × plant diversity | | | | −0.177 | | | | |
| Age × streetscape quality | | | | 0.003 | | | | |
| NR | | | | | 0.002 | 0.001 | | |
| NR × garden plant diversity | | | | | | 0.008 | | |
| NR × streetscape quality | | | | | | 0.000 | | |
| Gender | | | | | | | −0.100 * | −0.272 |
| Gender × plant diversity | | | | | | | | −0.297 |
| Gender × streetscape quality | | | | | | | | 0.509 ** |
| Adj. R² | 0.057 | 0.054 | 0.068 | 0.064 | 0.056 | 0.052 | 0.069 | 0.086 |
| ΔR² | 0.064 | 0.002 | 0.074 | 0.001 | 0.063 | 0.000 | 0.076 | 0.021 |
| ΔF | 9.44 *** | 0.429 | 11.19 *** | 0.227 | 9.31 *** | 0.063 | 11.27 *** | 4.71 ** |

*: $p < 0.05$, **: $p < 0.01$, *** $p < 0.001$. Dependent variable: GHQ (well-being change). N = 421 (Models 8a–c), 413 (Model 8d).

Age and gender appeared to be significant predictors for well-being in Step 1 of Models 8b and 8d, highlighting that they were relevant covariates (Table 6). However, in Step 2 of all the models, only the interaction term of gender and streetscape quality was a significant predictor, with streetscape quality itself no longer being significant, and the model overall improving. As described in the Section 2.3.2, the responses indicating "male" were coded as "0" and those indicating "female" as "1". Thus, the interaction term's β = 0.509 ($p < 0.01$) pertains to responses that were "female", while the main effect parameter, i.e., streetscape quality, which pertains to both "male" and "female", was no longer significant. This result indicated that that the quality of the streetscape greenery was associated with positive changes in well-being for women, but not for men.

The level of education, the age group, and the extent of nature connectedness were not significant moderators in the associations between streetscape greenery quality and garden plant diversity on the one hand, and changes in well-being on the other hand. Only gender moderated the relationship between streetscape greenery quality and changes in well-being.

## 4. Discussion

This study's findings explained some of the variation in the changes in mental well-being after the onset of the COVID-19 pandemic in 2020 by differences in access to different types of outdoor spaces and greenery in the residential environment. We identified which of the examined types and characteristics of green spaces and natural elements were most relevant and investigated the pathways in terms of exposure and engagement. The variables for exposure and engagement, as defined in the framework by Bratman et al. (2019), were tested as mediators in the pathway from green space to well-being benefit. The results are discussed in the following paragraphs, starting with those for the private gardens and then going on to neighbourhood greenery.

*4.1. Private Garden*

4.1.1. Characteristics

Out of the private outdoor spaces, the results pointed specifically to gardens, as being associated with a smaller negative (or even a positive) change in well-being. To examine which garden features were most relevant for well-being, we looked at size, overall greenness (i.e., abundance of vegetation), the abundance of specific types of vegetation, and the overall plant and bird diversity. Out of these, only the size and overall plant diversity remained significant. These findings are in line with existing research suggesting that perceived biodiversity is associated with well-being [41]. Although perceived biodiversity may not be reflective of biodiversity at the genetic or species level [41,42], we consider it to be important with respect to mental health. Further research could examine the links between different measures of biodiversity and mental health. Such knowledge would be useful for highlighting potential synergies between goals in the fields of health and ecology in urban land use governance. Studies show that planners are increasingly making gardens smaller or completely leaving them out of housing projects [43], and that they consider them nonessential [44]. Our findings and existing research suggest that gardens do have an important contribution to urban residents' mental health, especially during the pandemic lockdowns [45]. Thus, they should not be overlooked.

4.1.2. Exposure and Engagement as Mediators

For gardens, the frequency of contact (as exposure) and gardening (as engagement) were included as potential mediators. The results showed that neither was a mediator, suggesting that without visits to and engagement with the garden, the plant diversity and size were beneficially associated with changes in well-being. A possible explanation is that there may have been visual contact with the garden, which was sufficient to provide some benefit. Our results are not in line with existing research, showing that gardening mitigated stress during the pandemic in 2020 [46]. However, in this study, gardening as a form of engagement was operationalised in relation to the total time spent in the garden, which itself was not measured. Further research could revisit gardening as a mediator, for example, by looking at the total time spent engaging in it, and could also explore other, more specific interactions, such as taking photos, birdwatching, planting seeds, and feeding wildlife [47,48], to operationalise engagement with green spaces.

*4.2. Public Greenery*

4.2.1. Quantity or Quality, Type

In terms of public greenery, although initially the quantities of both public green areas and streetscape greenery were significantly associated with well-being, adding quality made them both redundant. Subsequently, the perceived streetscape greenery quality proved to be more strongly associated with well-being than the quality of public green areas, rendering the latter's predictive contribution insignificant. The high private garden ownership among respondents may have reduced the relevance of public green areas' quantity and quality by reducing the need for such areas. Moreover, streetscape greenery is something people likely come into contact with more often in their daily lives without purposeful visits (as would be the case for public green areas). The quality of the greenery in the street may also reflect the quality of the neighbourhood environment in general [16]. Further research could incorporate neighbourhood quality, examining its relationship with both streetscape greenery quality and mental health.

4.2.2. Exposure as a Mediator

The results did not indicate that the frequency of visits to a public green area was a mediator in the link between this area and well-being. This is not in line with previous research [27]. However, most survey respondents—despite the high proportion of private garden ownership—visited these green areas almost every day (35.9%) or a few times per week (30.3%). Thus, the insufficient variation in the frequency variable may account for

the lack of significant covariation with changes in well-being. Other than the frequency of contact, a relevant variable for measuring exposure is the time spent in a green area. Studies have shown a significant positive relationship between the time spent in green spaces in urban and peri-urban areas and mental health benefits [25,26]. Overall, the total time spent in green areas or the average duration per visit (alongside frequency), as well as different forms of engagement within these areas, may be useful additional mediators to test, both for private and public greenery.

### 4.3. Relative Importance of Different Types of Greenery

Lastly, a green view from within the home, when examined on its own, was associated with improved well-being, aligning with other studies during the pandemic [12,49]. This is an important finding because green window views may be even more beneficial in stricter lockdown situations than was the case in the Netherlands in 2020. A study comparing the importance of greenery for well-being during the lockdowns in Spain and Portugal suggested that, under a stricter lockdown, private greenery had a stronger beneficial association with mental well-being [50]. Our sample had high garden ownership, and green window views may also be more relevant for people who do not have a garden or other private outdoor area. Overall, a key finding of this research is that the streetscape greenery remained important even for people who have a garden. A garden's plant diversity (but not size) and the quality of streetscape greenery had the strongest association with well-being. This highlights the fact that it is important to focus on the quality and characteristics of green spaces, and not just on quantity. The results suggest, in line with other research [51], that private gardens and public green spaces are not interchangeable in their effects on urban dwellers—each comes with opportunities for different interactions with nature, and a different set of physical and social activities.

### 4.4. Moderation

Studies have suggested that there is no one type of green space which is the most health-promoting for everyone, everywhere, at all times [15]. The present research found that the quality of the streetscape greenery and garden plant diversity were beneficially associated with changes in well-being, regardless of a participant's age group, level of education, and nature connectedness. Only gender significantly moderated the relationship between the quality of streetscape greenery and well-being—higher quality was only important for women. Women may spend more time in their neighbourhood and on their street, as they are more likely to be caregivers and the ones conducting domestic activities such as grocery shopping [52]. This may have been the case during the pandemic too [53]. Additionally, scholars hypothesise that women and men use and perceive green spaces differently, in the way that women's use of green space is more strongly influenced by its quality and perceived safety [54,55]. Research on happiness—defined as "positive emotions expressing the achievement of the highest possible satisfaction with life" and correlated with aspects of well-being, health, and safety—during the pandemic in 2020 offers similar insights. It has indicated that the correlations between happiness and satisfaction with the quality of the residential context were stronger for women than men [56]. As further research is conducted on different types of and contact with nature and their effects on human well-being, it is important to acknowledge that differences between people may exist in these regards. It is necessary to understand these differences to be able to design urban areas which serve the health of all residents.

### 4.5. Limitations

A main limitation is that the study was prone to the common method bias, in which having a single source of information for the different variables can lead to overestimations in the strength of associations [57]. Self-reported changes in well-being after the onset of the COVID-19 pandemic is not the same as the actual impact of the pandemic and its associated measures on well-being. However, they are indicative in this regard, which still

makes them valuable for insights about those changes. Overcoming these limitations could be achieved through obtaining measures for the different variables from different sources (e.g., aerial imaging for quantities of greenery) or creating temporal (e.g., via longitudinal design) or proximal (e.g., by positioning measures of the same construct at least six items apart in a questionnaire) separation between measurements [58]. Due to the cross-sectional design of the study, one should exercise caution with regard to causal interpretations of the observed associations (the term 'effect' is used in this article in a statistical, predictive sense). Experimental designs, for example, could elaborate on findings such as the ones we present here. As our findings support associations, they are thus informative for further research on this topic, which can reduce the limitations of this study's design.

This study relied on the perceived quantity and quality of the natural elements and green areas, but many factors, including poor well-being, may affect such perceptions. As mentioned, the pandemic has had negative impacts on mental health, including causing higher stress. Increased stress has been associated with a higher negativity bias [59]. In this study, this could be expressed though, for example, participants giving lower scores for the quality of the public green greenery (e.g., seeing it as poorly maintained or lacking variety), thus leading to underestimations of its positive contributions to health. A negativity bias or higher anxiety may also increase experiences of biophobia, which could be an additional barrier to urban residents receiving the restorative benefits of nature [60]. This highlights the importance of studying perception, how it relates to reality, and how those relate to well-being, as well as the importance of examining variables that mediate or moderate these relationships.

The moderation analyses were exploratory, and with them, we also aimed to assess whether the non-representativeness of our sample on these characteristics is likely to have affected our outcomes strongly. The questionnaire was distributed in a way that it may have created a bias towards participants who have attained higher levels of education and are also more connected to nature. Indeed, the sample was not representative of the Dutch population in terms of education [61]: those with lower levels, and presumably a lower socioeconomic status, were underrepresented, as were men. The moderation analyses indicated that the latter was relevant for the representativeness of the outcome. The level and distribution of nature relatedness in the Dutch population are unknown. However, as with education, nature relatedness was not a significant moderator. This suggests that the sample (potentially) not being representative with regard to these characteristics is unlikely to have affected the outcomes of the regression analyses strongly.

The high prevalence of garden ownership was another potential bias in the sample, although reliable figures on this prevalence in the Netherlands at the level of individuals (rather than dwellings or households) are not available. However, other studies found that greenery in the residential environment had stronger associations with the health of more deprived communities and people with lower socioeconomic status (SES), and could mitigate inequalities in mental well-being [62,63]. This suggests that, in the case of an upward bias in the prevalence of garden presence due to low-SES people being underrepresented, the strength of associations is more likely to have been under- than overestimated. The findings of this study indicate that even for those who had a private garden, the quality of their streetscape greenery was still significantly beneficially associated with changes in well-being. Thus, our results put further emphasis on the important contributions of public greenery to mental health, when examined from the lens of garden owners too.

## 5. Conclusions

The relationship between public health and urban design has gained prominence with the onset of the COVID-19 pandemic, and continues to be relevant as cities grow in population and size worldwide. This work contributed to improving our understanding of the pathway between the residential environment and health. We investigated to what extent changes in the impact of COVID-19 on self-reported levels of well-being could be

explained by differences in access to and contact with different types of greenery in the residential environment. Aligning with and adding to existing literature on how nature supports and protects health, our results indicate that natural elements and green areas that are perceived to be diverse and well-maintained contribute to the well-being of urban populations, and that this was also the case during the COVID-19 pandemic lockdowns and measures in 2020.

Our findings have important implications for policymakers and planners in designing cities that promote healthy and resilient communities. The highlight of this study is that not only the quantity, but also—and even more so—the quality of the natural elements and green areas appear to be relevant for supporting well-being. Both public—in particular, streetscape—greenery and private outdoor areas—in particular, gardens—were associated with a less negative or even a positive change in well-being since the start of the pandemic. A key finding is that the streetscape greenery remained a significant predictor for well-being even for participants who had a garden. The results underscore the importance of conserving and/or creating attractive public green spaces.

The findings support associations and are thus informative for further research on this topic, which could address the limitations of this study. The continued examination of the pathways between nature and health furthers our understanding of how and which types of green areas and natural elements are the most relevant for health. This knowledge can be used in decision-making to advance resource-efficient and effective urban design, which promotes the health of all residents.

**Supplementary Materials:** The following are available online at https://doi.org/10.5281/zenodo.68 95120/, accessed on 24 July 2022.

**Author Contributions:** Conceptualization, R.S., S.d.V. and J.V.; methodology, S.d.V. and R.S.; formal analysis, R.S.; investigation, R.S.; validation, S.d.V.; resources, J.V. and S.d.V.; data curation, R.S.; writing—original draft preparation, R.S.; writing—review and editing, R.S., S.d.V. and J.V.; visualisation, R.S.; supervision, S.d.V. and J.V.; project administration, R.S. All authors have read and agreed to the published version of the manuscript.

**Funding:** The contribution of Sjerp de Vries was partially financed by the Foundation TKI Horticulture and Starting Materials (18042). This research was conducted for Ralitsa Shentova's master's thesis and received no other funding.

**Informed Consent Statement:** Informed consent was obtained from all subjects involved in the study.

**Data Availability Statement:** The data presented in this study are available upon request from the corresponding author.

**Acknowledgments:** We are grateful for Wichertje Bron's support to include an announcement about the survey in NatureToday's newsletter and to Lea Barbett for the fruitful discussions on the topics of this research.

**Conflicts of Interest:** The authors declare no conflict of interest.

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
