# Peer review of "Well-Being in the Time of Corona: Associations of Nearby Greenery with Mental Well-Being during COVID-19 in The Netherlands"

_sustainability, doi:10.3390/su141610256_

Round 1

Reviewer 1 Report

The paper is overall well-structured and well written. In some places language can be improved to increase clarity. The statistical analysis appears sound and is appropriate. 

There are two concerns with the study that cannot be retrospectively fixed but that need to be adequately considered during analysis (and mentioned in the limitations)

1) the sample population is rather homogenous with over 80% having access to a private garden. This was mentioned in the limitations. Additionally, you should consider that having a private garden may influence the use of public green spaces, especially during the pandemic. 

2) the adopted GHQ cannot really capture the change in well-being. The change of the word 'usually' to 'before' does frame the scoring in the pandemic, however, a cross-sectional study is not very well suited to capture change. Especially as the word change only applies in the negative extreme category. This is a limitation that should be mentioned more explicitly.

Additionally, it would be interesting to also have a table showing the GHQ broken up into its components

The conclusion can be a bit more focused on the key results of the study and could be improved.

Specific comments:

Lines 177 – 179 à It would be useful to mention the minimum and maximum scores possible to illustrate the reverse coding. I.e. a score of 4 implies ‘perfect’ wellbeing, while a score of 1 indicates the lowest wellbeing score.

Lines: 189 à Table 1: It would be useful to include in the first section ‘outdoor areas attached to home’  all participants, i.e. adding the option ‘no outdoor areas attached’. This would give a more complete picture of the sample population.

Line 212 à not necessary to emphasize that the log scale was used, this was clearly mentioned in the methods section

Line 222 à this wording is very strong and should be revised. I recommend not to emphasize the change since the pandemic so strongly. I suggest to write “… most important predictors of the change in mental well-being.”

Line 250 à here also I suggest to drop “after the onset of the COVID-19 pandemic”

Line 258 à here it may be reasonable to mention that the overall model fit is very low, pointing to confounding factors

Line 296 à just a comment. This may be related to the high proportion of people with private gardens. For people that may not have access to a private garden, green views may be more important. This could be a topic for further research

Line 349 à here it may be useful to consider the high proportion of private garden among the sample population. Having access to a private garden may influence the desirability of visiting public green spaces, especially during a pandemic and a lockdown. I do think that the bias in the sample can be stronger integrated into the interpretation of the results.

Line 361 à repetition, drop “, as was the case in Spain,”

Line 362-363 à Yes, this is a key finding! This can be a bit more prominent, maybe mention it also in the conclusion and maybe even the abstract.

Line 387 à this may also be related to the high proportions with gardens. It may be useful to examine the correlations between frequency of private garden use and frequency of public green space use. I would suspect that the effect of utilizing a private garden affects the effect of visiting public green spaces.

Line 390 à range not rage

Line 445 à here you may want to emphasize that the streetscape is still relevant even for people with private gardens.

Line 445-6 à “Their quantity mattered, but quality mattered even more for this beneficial association”, i.e. drop “when” [maybe consider revising this sentence]

Line 448 à “– to an extent – for whom”, I somewhat disagree and would suggest to drop this. As your sample is not representative and is actually quite biased towards higher educated people with private gardens, I am not sure that the study provides insight into what types of greenspace are most relevant for different groups. For people that have private gardens, of course that private garden is the most important, but what about people without private gardens? What type is most important for them?

Author Response

Dear Reviewer,

We appreciate the time and effort you have dedicated for feedback on our manuscript and are grateful for your detailed comments! They helped us to implement valuable improvements to our manuscript. We aimed to better address the major limitations that you highlighted by revising the relevant sections in our Discussion, as well as adding more detail to the Method section. In particular:

  • 1) the sample population is rather homogenous with over 80% having access to a private garden. This was mentioned in the limitations. Additionally, you should consider that having a private garden may influence the use of public green spaces, especially during the pandemic.”
    • This was taken better into account in the Discussion, acknowledging that the high private garden ownership among respondents may influence the strength of the effect of public green areas’ quantity and quality by reducing the need for such areas.
  • 2) the adopted GHQ cannot really capture the change in well-being. The change of the word 'usually' to 'before' does frame the scoring in the pandemic, however, a cross-sectional study is not very well suited to capture change. Especially as the word change only applies in the negative extreme category. This is a limitation that should be mentioned more explicitly.”
    • More information on the changes we implemented in the wording of GHQ was included in Method - word changes appear in the main question, as well as in all response options; the limitations of the cross-sectional design were further underscored, suggestions for specific ways to address them in further research were included.
  • Additionally, it would be interesting to also have a table showing the GHQ broken up into its components”
    • We clarified in the Method section that GHQ-12 is meant to be used as a unidimensional measure for psychological distress, as cited in literature; for this reason, its components were not specified. However, the entire questionnaire will become available in Supplementary Materials where all questions are visible.
  • The conclusion can be a bit more focused on the key results of the study and could be improved.”
    • The Conclusion has been revised as suggested.

We incorporated all your other comments by implementing the following changes in the manuscript:

  • Lines 177 – 179 à It would be useful to mention the minimum and maximum scores possible to illustrate the reverse coding. I.e. a score of 4 implies ‘perfect’ wellbeing, while a score of 1 indicates the lowest wellbeing score.
    • A sentence was added on this: “Thus, a score of 1 indicates maximally worsened well-being, and a score of 4 – most improvement in well-being.”
  • Lines: 189 à Table 1: It would be useful to include in the first section ‘outdoor areas attached to home’ all participants, i.e. adding the option ‘no outdoor areas attached’. This would give a more complete picture of the sample population.
    • The table was revised accordingly.
  • Line 258 à here it may be reasonable to mention that the overall model fit is very low, pointing to confounding factors
    • A sentence was added on this.
  • Line 296 à just a comment. This may be related to the high proportion of people with private gardens. For people that may not have access to a private garden, green views may be more important. This could be a topic for further research
    • This is a very relevant comment and was mentioned in the Discussion section.
  • Line 349 à here it may be useful to consider the high proportion of private garden among the sample population. Having access to a private garden may influence the desirability of visiting public green spaces, especially during a pandemic and a lockdown. I do think that the bias in the sample can be stronger integrated into the interpretation of the results.
    • Results showed that most respondents visited public green areas very frequently, despite most having a private garden, so we believe it is less likely that it affected the desirability to visit such spaces. However, we did elaborate in the Discussion that the high garden ownership may influence the strength of the effect of public green areas’ quality and quantity.
  • Line 362-363 à Yes, this is a key finding! This can be a bit more prominent, maybe mention it also in the conclusion and maybe even the abstract.
    • Thank you! We indeed included it more in the revised Conclusion and Abstract.
  • Line 387 à this may also be related to the high proportions with gardens. It may be useful to examine the correlations between frequency of private garden use and frequency of public green space use. I would suspect that the effect of utilizing a private garden affects the effect of visiting public green spaces.
    • As suggested, we did the analysis and the frequency of private garden use and public green space use were not correlated. We made sure to include this result in the Supplementary Materials (as we state in Method that additional analyses can be found there). The lack of effect, as we elaborated in the text, may be due to the insufficient variation in the responses (most people visited public green areas very frequently).

Writing was revised as suggested all the other comments, i.e.:

  • Line 212 à not necessary to emphasize that the log scale was used, this was clearly mentioned in the methods section
  • Line 222 à this wording is very strong and should be revised. I recommend not to emphasize the change since the pandemic so strongly. I suggest to write “… most important predictors of the change in mental well-being.”
  • Line 250 à here also I suggest to drop “after the onset of the COVID-19 pandemic”
  • Line 361 à repetition, drop “, as was the case in Spain,”
  • Line 390 à range not rage
  • Line 445 à here you may want to emphasize that the streetscape is still relevant even for people with private gardens.
  • Line 445-6 à “Their quantity mattered, but quality mattered even more for this beneficial association”, i.e. drop “when” [maybe consider revising this sentence]
  • Line 448 à “– to an extent – for whom”, I somewhat disagree and would suggest to drop this. As your sample is not representative and is actually quite biased towards higher educated people with private gardens, I am not sure that the study provides insight into what types of greenspace are most relevant for different groups. For people that have private gardens, of course that private garden is the most important, but what about people without private gardens? What type is most important for them?

Thank you again for your generosity in helping us improve our manuscript.

Kind regards,

The Authors

Reviewer 2 Report

The present manuscript reports data examining the associations between various aspects of greenspace and perceived changes in psychological well-being during the COVID-19 pandemic. A community sample (N=521) of individuals in the Netherlands was recruited and cross-sectional self-report data were collected. The manuscript is well written and addresses a timely and important topic. I believe that it is critical for the field to begin to explore the specific aspects of green space exposure that are relevant for health and mental health outcomes. Nonetheless, there are several issues that dampen my enthusiasm for the research.

(1) The reliable and valid measurement of greenery is crucial for this study, but little specific information is provided about the self-report measure that assessed these variables. Consequently, it is difficult to evaluate the results of this study.

It is unclear what specific variables were assessed with the measure, how the questions were worded, or what kind of response scale was used. How was the "garden's greenness," or the "abundance of grass, flowers/herbs, shrubs, and trees," or "bird diversity" assessed? How would a participant who rarely spends time outside be able to provide valid ratings on the diversity of birds, plants etc? How were participants asked to record the size of their gardens and do we have evidence that such estimates of garden size are valid? To what extent would poor psychological well-being (which manifests itself in large part as depression and anxiety that involve negative cognitive biases in interpretation and perception) potentially bias ratings of the quantify and quality of green space?

(2) The study appeared to lack a coherent analytic strategy. For example, the logic of the regression models, particularly what variables were added at what steps, was not clear. Likewise, in the case of the moderation analyses, the authors stated that "multiplicative interaction terms were created between the most relevant predictors for well-being and education, nature relatedness, gender, and age" (p. 4). However, "most relevant" was not defined. Did this mean the variables with the largest effect sizes or lowest p values? If so, what was the reasoning? A statistical interaction (moderation) can occur even with variables that have weak or non-significant main effects.

(3) The study relied on Baron and Kenny's (1986) approach to testing mediation, which is considered outdated. Instead, the emphasis in contemporary approaches is testing the significance of the indirect effect using the Sobel test or better yet bootstrapping (e.g. Hayes, 2009).

(4) In testing moderation, the authors dichotomized continuous measures. This reduces statistical power and is considered poor practice.  The authors might find Aiken and West's (1991) text "Multiple Regression: Testing and Interpreting Interactions" a useful starting place for guidance on these kinds of analyses.

(5) Supplementary materials referred to in text were not available at the url listed.

(6) The internal consistencies of the six-item Nature Relatedness Scale and the GHQ should be reported.

Reference

Hayes, A.F. (2009). Beyond Baron and Kenny: Statistical Mediation Analysis in the New Millennium. Communication Monographs, 76, 408 - 420.

Author Response

Dear Reviewer,

We appreciate the time and effort you have dedicated for feedback on our manuscript and are grateful for your detailed comments. They helped us to implement valuable improvements to our manuscript and research. We have incorporated your suggestions, largely through major revisions to our Method. Below, we offer detailed responses on the changes we implemented to address your concerns.

  • The reliable and valid measurement of greenery is crucial for this study, but little specific information is provided about the self-report measure that assessed these variables. Consequently, it is difficult to evaluate the results of this study. It is unclear what specific variables were assessed with the measure, how the questions were worded, or what kind of response scale was used. How was the "garden's greenness," or the "abundance of grass, flowers/herbs, shrubs, and trees," or "bird diversity" assessed? How would a participant who rarely spends time outside be able to provide valid ratings on the diversity of birds, plants etc? How were participants asked to record the size of their gardens and do we have evidence that such estimates of garden size are valid? To what extent would poor psychological well-being (which manifests itself in large part as depression and anxiety that involve negative cognitive biases in interpretation and perception) potentially bias ratings of the quantify and quality of green space?
    • As a start, in the Method section, we included more examples of the exact questions and response options we used and which variables (e.g. greenness, bird diversity) we refer to with them. We also elaborated that for garden size, participants were asked to provide an estimate in square meters (Lines 127-148).
    • With regards to the validity of these measurements, we agree that perceptions are not always accurate with regard to the objective state of affairs and that, thus, perceptions of those variables may not reflect reality. However, perceptions are relevant as such, research indicating that perceived biodiversity is potentially more important with regard to mental health than actual biodiversity, and that it indeed does not reflect actual biodiversity (Dallimer et al., 2012). In this study, perception is what we focused on, without an attempt to assess what this entails in objective terms of (the natural part of) the physical environment. Taking into account your comment, we revised the manuscript, implementing changes in the text (at the end of the Introduction, in Discussion, and Conclusions) to acknowledge that perception and reality refer to different constructs, requiring different measures, and to clarify what our aim is.
    • Your comment on poor psychological well-being potentially biasing the ratings is also relevant. To address this better, we included a new paragraph on the impacts of the negativity bias in our study (Lines 478-488).
    • As the study is cross-sectional, the causality of observed associations is always an issue. In the revision, we made sure to acknowledge more clearly, elaborating on the Limitations sub-section, as well as suggesting ways to overcome our limitations in further research.
  • The logic of the regression models, particularly what variables were added at what steps, was not clear. Likewise, in the case of the moderation analyses, the authors stated that "multiplicative interaction terms were created between the most relevant predictors for well-being and education, nature relatedness, gender, and age" (p. 4). However, "most relevant" was not defined.
    • To address the issues flagged here, more detail was added in a section “Statistical Analyses”, explaining that logic of the regression models was “first adding the more general characteristics, i.e. how much greenery is present, then the more specific ones in the following steps, i.e. what is that greenery like” (Lines 172-4)
    • Later on, “most relevant” is defined: “The predictors that remain significant in the last steps of the regression analyses are considered “most relevant” for well-being in this study and are used in the following analyses for mediation and moderation.” We followed this approach as it was aligned with the framework by Bratman et al. (2019) on the route from nature to mental health. Thus, in this study, the focus was on identifying the direct links between the natural features/types of greenery and well-being. Then, the moderation and mediation analyses acted as follow-up analyses to explain that link.
  • The study relied on Baron and Kenny's (1986) approach to testing mediation, which is considered outdated. Instead, the emphasis in contemporary approaches is testing the significance of the indirect effect using the Sobel test or better yet bootstrapping (e.g. Hayes, 2009).
    • As suggested, we conducted bootstrapping analyses to provide improved estimations for the indirect effects.
  • In testing moderation, the authors dichotomized continuous measures. This reduces statistical power and is considered poor practice.
    • Out of the four moderators, this loss of information mostly pertains to nature connectedness which is a continuous measure. In our revision, we address this issue more explicitly in the Discussion, acknowledging the loss of information and citing research on the topic. We conducted the moderation analyses as exploratory, crude analyses, and to examine whether the non-representativeness of our sample on these aspects is likely to have strongly affected our outcomes.
  • Supplementary materials referred to in text were not available at the url listed.
    • The correct link to the Supplementary Materials was added.
  • The internal consistencies of the six-item Nature Relatedness Scale and the GHQ should be reported.
    • Cronbach’s alpha was reported for both.

Thank you again for your generosity in helping us improve our manuscript.

Kind regards,

The Authors

References:

Bratman, G. N., Anderson, C. B., Berman, M. G., Cochran, B., De Vries, S., Flanders, J., ... & Daily, G. C. (2019). Nature and mental health: An ecosystem service perspective. Science advances, 5(7). https://doi.org/10.1126/sciadv.aax0903

Dallimer, M., Irvine, K. N., Skinner, A. M., Davies, Z. G., Rouquette, J. R., Maltby, L. L., ... & Gaston, K. J. (2012). Biodiversity and the feel-good factor: understanding associations between self-reported human well-being and species richness. BioScience, 62(1), 47-55. https://doi.org/10.1525/bio.2012.62.1.9

Reviewer 3 Report

Dear Authors,

I found your work very interesting. The idea set behind the text meets the scope of the Journal as the article refers to the quality of green infrastructure and its impact on residents' well-being. The article's structure is clear and coherent. The article is well referenced. Although, the topic is not new, the results bring some fresh insight into the role of the diversity of urban plantings on residents' health. I believe it will be a valuable contribution for the Journal.

best regards,

the reviewer

Author Response

Dear Reviewer,

We appreciate the time you have dedicated to reading our manuscript and are grateful for your comments!

Kind regards,

The Authors

Reviewer 4 Report

Considering the extraordinary event of the pandemic that has changed and is changing the way we work (e.g. working from home), our social interactions (e.g. social distancing), and the way we experience common spaces (e.g. banning going out, going to the park or aggregating) enhancing green spaces and the benefits for individual health must be a duty of social and urban policies. The work aimed to examine the link between having a private garden, a green view from the window, the perceived amount and quality of neighborhood green areas and streetscape greenery, and self-reported change in mental well-being since the onset of the pandemic. Results showed that gardens with diverse plants, as well as well-maintained, attractive, and varied streetscape greenery, were strongly associated with less negative changes in well-being. Since the positive effect of natural elements (parks, gardens, or window views) on individual health is already known in the literature, I would suggest that the authors emphasize the novelty of the results and the original aspects of the work. 

Author Response

Dear Reviewer,

We appreciate the time and effort you have dedicated for feedback on our manuscript and are grateful for your detailed comments! They helped us to implement valuable improvements to our manuscript and research. We incorporated your suggestions by highlighting the contribution to literature and practice, largely through rewriting the Conclusion and editing the Discussion and Abstract.

Below, we offer detailed responses on the changes we implemented to address your comments.

  • I would also suggest adding details to section Study design and dissemination (p. 3). In particular, it would help the reading and understanding of the work to add examples of a few questions that the subjects had to answer using the 7-point Likert scale ( which labels were at the endpoints?). Also, when did participants complete GHQ-12? before or after the survey?
    • The Methods section was rearranged into new sub-sections, with more detail added to them. We now start by explaining the order of topics, as experienced by the survey respondents. Sample questions and their endpoints were added for a few main variables.
  • I would also suggest improving the structure of the section Statistical analysis and Results (p. 3 and p. 5). To aid understanding, I would again specify the variables taken into account for the regression. In addition, I would organise the Results section with a sub-section for the t-tests performed.
    • As suggested, the Results section was organised in sub-sections, which were kept the same for the Discussion.

Additional references were cited on the following occasions, as suggested:

  • In the introduction (p. 1 lines 34,35) the author(s) say "However, research also suggests that more greenery in the residential environment may mitigate these negative mental health impacts". Cite some evidence.
  • In the introduction (p. 2, line 62) the following is reported: “Studies show that a green window view is also associated with well-being”. However, only one work was cited against the studies mentioned.
  • In the section Covariates (p. 3, line 137) the author(s) state “Socio-demographic and other variables known to affect well-being and outdoor space use were also measured”. Cite some evidence.

Thank you again for your generosity in helping us improve our manuscript.

Kind regards,

The Authors

Reviewer 5 Report

The paper aims to explain changes in mental well-being after the COVID 19 pandemic in 2020 by differences in access to different types of outdoor spaces and greenery in the residential environment.

The introduction provides a generalised background to understand the topic. However, the article could be implemented in the structure to clarify the methodology adopted and the results achieved.

-       I suggest paragraph 2. Materials and Methods be more clearly defined. 

In my opinion, the following sub-sections could be inserted:  

“Sample” should include the part related to Participants (line 183) and Settings (line 101). In particular, I suggest anticipating the limits regarding the sample of participants (line 428) and specifying the urban and building characteristics of the reference context.

It is important to emphasise that the sample is not representative so that the results, which cannot be generalised, can be adequately evaluated.

“Questionnaire Design” should include Study design and disseminationMental well-being assessment, Covariates.

“Data Analyses Methodology” should include Statistical analyses, Data preparation. 

-       I also suggest keeping the sub-paragraphs the same in the 3. Results and 4. Discussion so that it is more explicit.

-       I would include a discussion of limitations in the conclusions, also suggesting ways to overcome them.

Other comments on the paper:

- I suggest rephrasing the first sentence of the abstract, pointing out that there is already much scientific literature on the subject, but little data on the pandemic period in particular (line 10)

- Specify the framework in which the research is carried out, even if it is not funded; for example, from whom does the research question emerge? Which partners are involved? (line 93)

-       Please insert the email of the third author

Author Response

Dear Reviewer,

We appreciate the time and effort you have dedicated to providing feedback on our manuscript and are grateful for your comments! They helped us to implement valuable improvements to our manuscript. Overall, we incorporated your suggestions by revising the structure of the Methods, Results, and Discussion sections, creating sub-sections and keeping them the same to ensure there is continuity and clarity. Below, we offer our detailed responses to your comments.

  • I suggest paragraph 2. Materials and Methods be more clearly defined. In my opinion, the following sub-sections could be inserted: “Sample” should include the part related to Participants (line 183) and Settings (line 101).
    • In this case, we chose to keep the current structure, starting with the Setting, as that creates a frame for the Questionnaire Design (explaining why it was in Dutch as well as English), and ending with Participants as the tables presented there include information on variables described earlier in the Questionnaire Design. Thus, the reader can be introduced to context, the relevant variables, then the results on those variables, and that can create a smoother transition to the Results section.
  • In particular, I suggest anticipating the limits regarding the sample of participants (line 428) and specifying the urban and building characteristics of the reference context. It is important to emphasise that the sample is not representative so that the results, which cannot be generalised, can be adequately evaluated.
    • To address this comment, we stressed that it is a convenience sample, depending on self-enrolment of participants, in the Methods section, where we talk about the Questionnaire Design and Dissemination. By looking into potential moderators, we assess to some extent the consequences of certain ways in which the sample is not representative of the Dutch population. However, in the revised manuscript, we also address the non-representativeness in more detail in the Discussion.
  • “Questionnaire Design” should include Study design and dissemination, Mental well-being assessment, Covariates.
    • Taking into account the suggestion and after experimentation with the order and structure, a “Study Design” section was created with the sub-sections “Residential Greenery” (including more detail on the questions asked), “Well-being Assessment”, “Covariates”, and “Dissemination”.
  • “Data Analyses Methodology” should include Statistical analyses, Data preparation.
    • As suggested, a “Data Analysis” section was created with “Statistical Analyses” and “Data Preparation” sub-sections.
  • I also suggest keeping the sub-paragraphs the same in the 3. Results and 4. Discussion so that it is more explicit.
    • This was implemented as suggested.
  • I would include a discussion of limitations in the conclusions, also suggesting ways to overcome them.
    • We took this into account by elaborating how to overcome the limitations, citing relevant literature in the Discussion section. The Conclusions section was revised, and also mentioned the importance of conducting further research on this topic, which addresses the limitations of our study.
  • I suggest rephrasing the first sentence of the abstract, pointing out that there is already much scientific literature on the subject, but little data on the pandemic period in particular (line 10)
    • Following this suggestion, we revised the first sentence of the abstract to highlight that there is little data on the specific types and characteristics that are relevant for mental health and that is particularly the case for research during the pandemic.
  • Specify the framework in which the research is carried out, even if it is not funded; for example, from whom does the research question emerge? Which partners are involved? (line 93)
    • To address this comment, under “Funding”, we specified that this research was conducted as part of a master thesis (in addition to the fact that it received no external funding).
  • Please insert the email of the third author
    • We made sure to do this for the revised version too.

Thank you again for your generosity in helping us improve our manuscript.

Kind regards,

The Authors

Round 2

Reviewer 2 Report

I believe that this revised manuscript is stronger and more clear than the initial submission, but that there are remaining issues.

(1) I would still advise against dichotomizing continuous variables for the moderation analyses. Although the authors state this was done to simplify the analyses, there really is nothing gained in terms of simplicity or otherwise. Instead, there is loss of information and consequently a reduction in statistical power. The fact that these were exploratory analyses, as stated in the Discussion, is not a justification.

(2) The authors report a statistically significant Sex x Streetscape greenery interaction, and state that this indicated that women were more impacted than men. However, the authors failed to report simple slope analyses to unpack the form of this interaction. Consequently, there is no statistical evidence provided to support the claim that the effect was stronger for females.

(3) Only 3% of the sample did not have access to some form of green space at their place of residence, which calls into question whether or not the significant difference in well-being between those and without home-based green space will replicate in other samples.

(4) I am not following the logic of the mediation analyses that were tested on page 7. Why was quality of public greenspace tested as an IV, but not quality of streetscape greenery? The previous analyses in Table 3 suggested that the latter was even more important than the former in well-being.

(5) On page 10 the authors state "Results showed that the frequency of contact was a partial mediator..."  However, this statement is contradicted by the results reported on page 7, which indicate that the indirect effect was not statistically significant. In other words, there was no support for either partial or full mediation.

(6) There are some writing issues in the discussion of the data analytic plan on page 5.

-- "Independent samples t-tests were used to examine significant differences in well-being..." Remove the word "significant" from this sentence.

-- "Correlations were used to examine if there were significant relationships between variables. Linear regression analyses were used to study the predictive power of the independent variables and to quantify their relationship with the dependent variable (change in well-being)." These sentences strike me as vague and do not clearly distinguish the purpose of the correlational and regression analyses.

-- "GHQ-12 scores were reverse-coded so higher scores illustrate a less negative or even slightly positive change in well-being..." Remove the word "slightly" as it implies that stronger positive changes would not have also been represented by higher scores.

(7) Page 11: "Studies suggest that there is no one type of green space which is the most health-promoting for everyone everywhere at all times [15]. This research found that the quality of the streetscape greenery and garden plant diversity..."  Change "This research..." to "The present research..."

Author Response

Dear Reviewer,

We are again grateful for your time and valuable suggestions for improving the quality of our manuscript. We have taken them into account, and offer our responses to each comment below.

(1) I would still advise against dichotomizing continuous variables for the moderation analyses. Although the authors state this was done to simplify the analyses, there really is nothing gained in terms of simplicity or otherwise. Instead, there is loss of information and consequently a reduction in statistical power. The fact that these were exploratory analyses, as stated in the Discussion, is not a justification.

  • Thank you for highlighting this again. We have modified our approach to the moderation analysis with the continuous variable (NR, nature relatedness) accordingly. We mean-centered it, as well as the relevant predictor variables, before creating the interaction terms, and then ran the regression with them (Lines 213-16). The results were not significant. In the Discussion, limitations section (Lines 502-8), we added some lines to make clearer what the moderation analyses' results indicate regarding the likely influence of the non-representativeness of the sample on the representativeness of the outcomes of the regression analyses.

(2) The authors report a statistically significant Sex x Streetscape greenery interaction, and state that this indicated that women were more impacted than men. However, the authors failed to report simple slope analyses to unpack the form of this interaction. Consequently, there is no statistical evidence provided to support the claim that the effect was stronger for females.

  • We agree that the description of the Gender x Streetscape greenery moderation in the Results section was limited. We elaborated on the description of the results (Lines 353-8). We repeated that men were coded as 0 and women as 1. Thus, we highlighted that this means the reported Beta-value in Table 6 (β = .509, p < .01) already indicates that the association between streetscape quality and GHQ-change is stronger/more positive for women. The result that the main effect parameter, i.e. Streetscape quality (β = -.038), which pertains to both men and women, was not significant. This indicates that for men there is no longer a significant positive association. With these additional explanations, we believe a simple slope analysis is not necessary as the interpretation of the regression parameter for the interaction in this case is more straightforward.

(3) Only 3% of the sample did not have access to some form of green space at their place of residence, which calls into question whether or not the significant difference in well-being between those and without home-based green space will replicate in other samples.

  • Thank you for this observation. The 3% pertains to people without an outdoor space, which need not necessarily be green (see Table 1 on page 5). With regard to the high percentage of garden ownership, we already acknowledge this in the Discussion and suggest that present results may be on the conservative side - existing research shows that associations with public green space are likely to be stronger for those without home-based green space.

(4) I am not following the logic of the mediation analyses that were tested on page 7. Why was quality of public greenspace tested as an IV, but not quality of streetscape greenery? The previous analyses in Table 3 suggested that the latter was even more important than the former in well-being.

  • In this case, the frequency referred only to visits to public green areas, whereas people come into contact with their streetscape greenery each time they leave the house. Therefore, we did not consider this visit frequency as a mediator in relation to streetscape greenery. This mediation analysis still followed the logic of the previous ones: the quality of public green areas was more significant than the quantity, so we again chose the more relevant predictor from those relating to public green areas.

(5) On page 10 the authors state "Results showed that the frequency of contact was a partial mediator..."  However, this statement is contradicted by the results reported on page 7, which indicate that the indirect effect was not statistically significant. In other words, there was no support for either partial or full mediation.

  • Thank you for pointing this out, there was indeed no evidence for mediation in the results. We have modified the text accordingly.

(6) There are some writing issues in the discussion of the data analytic plan on page 5.

-- "Independent samples t-tests were used to examine significant differences in well-being..." Remove the word "significant" from this sentence.

  • We agree with this and have incorporated your suggestion.

-- "Correlations were used to examine if there were significant relationships between variables. Linear regression analyses were used to study the predictive power of the independent variables and to quantify their relationship with the dependent variable (change in well-being)." These sentences strike me as vague and do not clearly distinguish the purpose of the correlational and regression analyses.

  • We modified the sentences to be clearer: “Correlations were used to examine if there were significant relationships between the different independent variables (IVs)” (we do this in Table 4 on page 8) and “Linear regression analyses were used to study the predictive power of the various IVs and to quantify their relationship with the dependent variable (change in well-being).”

-- "GHQ-12 scores were reverse-coded so higher scores illustrate a less negative or even slightly positive change in well-being..." Remove the word "slightly" as it implies that stronger positive changes would not have also been represented by higher scores.

  • Thank you for this suggestion. We agree and have revised the text (Line 221). 

(7) Page 11: "Studies suggest that there is no one type of green space which is the most health-promoting for everyone everywhere at all times [15]. This research found that the quality of the streetscape greenery and garden plant diversity..."  Change "This research..." to "The present research..."

  • We revised the text accordingly (Line 450). 

Kind regards,

The Authors

Reviewer 5 Report

Dear authors,
thank you for the detailed feedback on the revisions I suggested.
The manuscript has been sufficiently improved

Author Response

Dear Reviewer, 

Thank you again for your comments and helping improve our paper.

Kind regards,

The Authors